# Metabolic modulation regulates cardiac wall morphogenesis in zebrafish

Ryuichi Fukuda[1]*, Alla Aharonov[2], Yu Ting Ong[3], Oliver A Stone[1†], Mohamed El-Brolosy[1], Hans-Martin Maischein[1], Michael Potente[3], Eldad Tzahor[2], Didier YR Stainier[1]*

[1]Department of Developmental Genetics, Max Planck Institute for Heart and Lung Research, Ludwigstrasse, Germany; [2]Department of Molecular Cell Biology, Weizmann Institute of Science, Rehovot, Israel; [3]Angiogenesis & Metabolism Laboratory, Max Planck Institute for Heart and Lung Research, Ludwigstrasse, Germany

**Abstract** During cardiac development, cardiomyocytes form complex inner wall structures called trabeculae. Despite significant investigation into this process, the potential role of metabolism has not been addressed. Using single cell resolution imaging in zebrafish, we find that cardiomyocytes seeding the trabecular layer actively change their shape while compact layer cardiomyocytes remain static. We show that Erbb2 signaling, which is required for trabeculation, activates glycolysis to support changes in cardiomyocyte shape and behavior. Pharmacological inhibition of glycolysis impairs cardiac trabeculation, and cardiomyocyte-specific loss- and gain-of-function manipulations of glycolysis decrease and increase trabeculation, respectively. In addition, loss of the glycolytic enzyme pyruvate kinase M2 impairs trabeculation. Experiments with rat neonatal cardiomyocytes in culture further support these observations. Our findings reveal new roles for glycolysis in regulating cardiomyocyte behavior during cardiac wall morphogenesis.

*For correspondence:
Ryuichi.Fukuda@mpi-bn.mpg.de
(RF);
Didier.Stainier@mpi-bn.mpg.de
(DYRS)

Present address: †Department
of Physiology, Anatomy and
Genetics, BHF Centre of
Research Excellence, University
of Oxford, Oxford, United
Kingdom

Competing interest: See
page 10

Reviewing editor: Marianne E
Bronner, California Institute of
Technology, United States

## Introduction

During development, the heart undergoes a series of morphogenetic changes to form a functional cardiac wall structure (*Moorman and Christoffels, 2003*; *Staudt and Stainier, 2012*). The outer wall of the developing ventricle consists of compact layer cardiomyocytes (CMs), while the inner wall consists of complex muscular ridges, termed trabeculae, which facilitate efficient cardiac contraction and oxygenation of the cardiac wall prior to the formation of coronary vessels (*Sedmera et al., 2000*; *Staudt and Stainier, 2012*). Disruption of ventricular wall morphogenesis is associated with congenital cardiac malformations, the most common type of birth defects (*Fahed et al., 2013*), yet the cellular and molecular mechanisms regulating this complex process remain unclear.

Neuregulin (NRG)/Erb-b2 receptor tyrosine kinase (ERBB) 2/4 signaling has been shown to be essential for cardiac trabeculation. *Nrg1*, *Erbb2* and *Erbb4* knockout mice (*Gassmann et al., 1995*; *Lee et al., 1995*; *Meyer and Birchmeier, 1995*) and *nrg2a* (*Rasouli and Stainier, 2017*) and *erbb2* (*Liu et al., 2010*) mutant fish fail to form trabeculae. ERBBs are members of the epidermal growth factor (EGF) receptor tyrosine kinase family. NRGs are expressed by the endocardium (*Corfas et al., 1995*; *Meyer and Birchmeier, 1995*; *Grego-Bessa et al., 2007*; *Rasouli and Stainier, 2017*) and bind to ERBBs on CMs, triggering homo- or heterodimerization of ERBB family members and leading to activation of downstream pathways (*Sanchez-Soria and Camenisch, 2010*). However, the targets of ERBB2 signaling that regulate CM behavior during trabeculation have not been identified.

Cardiac metabolism has been extensively studied in adult animals due to its central role in supplying energy for cardiac contraction (*Doenst et al., 2013*; *Kolwicz et al., 2013*). Adult CMs rely mostly on fatty acids as an energy substrate, and they are oxidized in mitochondria to generate ATP

(*Ellen Kreipke et al., 2016*). Under conditions of hypertrophic or ischemic stress, CMs revert to glycolytic metabolism (*Doenst et al., 2013*), which is characteristic of embryonic cardiomyocytes and uses glucose as a fuel. Besides its role in energy generation, little is known about the role of metabolism during cardiac development.

Here, using high-resolution single cell imaging in zebrafish, we first show that developing CMs undergo extensive shape changes during the formation of the trabecular layer. By modulating glucose metabolism pharmacologically, we show that glycolysis regulates these processes. Using CM-specific loss- and gain-of-function models as well as mutant animals compromised in their glycolytic activity, we identify a role for glycolysis in cardiac wall morphogenesis. This study provides new insights into the role of cardiac metabolism in cardiac development.

## Results

### Cardiomyocytes that enter the trabecular layer exhibit distinct behaviors

During cardiac trabeculation in zebrafish and mouse, CMs delaminate from the compact layer to seed the trabecular layer (*Liu et al., 2010*; *Zhang et al., 2013*; *Staudt et al., 2014*; *Jiménez-Amilburu et al., 2016*; *Del Monte-Nieto et al., 2018*). Although CM behavior during trabeculation has been observed in zebrafish (*Staudt et al., 2014*; *Cherian et al., 2016*), the 3D morphology of single cardiomyocytes during the trabeculation process needs to be further explored. To this end, we performed 3D time-course imaging using chimeric hearts generated by cell transplantation. To label CM membranes and nuclei with EGFP and DsRed2 respectively, we used *Tg(myl7:EGFP-HRAS); Tg(myl7:nDsRed2)* cells as donors (*Figure 1a* and *Figure 1—figure supplement 1a*). We found that delaminating CMs exhibit morphological changes as well as rearrangements of contact sites (*Figure 1b–c"* and *Figure 1—figure supplement 1b–d*; *Figure 1—videos 1* and *2*), while CMs remaining in the compact layer do not exhibit such changes (*Figure 1d–e"*). To examine cell-cell junctions during delamination, we analyzed N-cadherin (Cdh2), a major adherens junction component, at single cell resolution, and to this end used *Tg(myl7:EGFP-HRAS); Tg(myl7:cdh2-tdTomato)* cells as donors (*Figure 1—figure supplement 1e*). We observed that N-cadherin localizes to protruding membranes in delaminating CMs (*Figure 1—figure supplement 1f–g"*) and to the lateral membranes of compact layer CMs (*Figure 1—figure supplement 1h–i"*), in agreement with a previous report (*Cherian et al., 2016*). Next, we analyzed sarcomere structure during delamination using *Tg(myl7:LIFEACT-GFP); Tg(myl7:nDsRed2)* cells as donors. We found that CMs display partial sarcomere disassembly in their protrusions when entering the trabecular layer (*Figure 1f–g"*). These data indicate that delaminating CMs exhibit distinct behaviors including dynamic cell shape changes.

### ERBB2 signaling activates glycolysis in cardiomyocytes

To gain additional insight into the molecular mechanisms that regulate trabeculation, we focused on ERBB2 signaling which is essential for this process in both mouse (*Gassmann et al., 1995*; *Lee et al., 1995*; *Meyer and Birchmeier, 1995*) and zebrafish (*Liu et al., 2010*; *Peshkovsky et al., 2011*; *Rasouli and Stainier, 2017*). In order to identify targets of ERBB2 signaling, we analyzed protein expression in a CM-specific transgenic mouse model inducibly expressing a constitutively active form of ERBB2 (CAERBB2) (*D'Uva et al., 2015*). We found that upon CAERBB2 overexpression (OE), several glycolytic enzymes were upregulated, while mitochondrial proteins and oxidative phosphorylation (OXPHOS)-related enzymes were downregulated (*Figure 2—source data 1*). We also tested the effects of *Erbb2* OE, which like *CAErbb2* OE, activates downstream signaling (*Pedersen et al., 2009*), on the expression of glycolytic enzyme genes in rat neonatal CMs, and found that many of them were upregulated (*Figure 2a*). Notably, we found that *Erbb2* OE in rat neonatal CMs greatly upregulated the levels of pyruvate kinase M2 (PKM2), a key glycolytic enzyme, (*Figure 2b and c*), and increased glycolytic activity as evidenced by measuring extracellular acidification rate (ECAR) (*Figure 2d*). NRG1 stimulation also activated glycolysis in rat neonatal CMs (*Figure 2—figure supplement 1a*). These findings are also supported by a study analyzing changes in mRNA levels in caErbb2 OE mouse hearts (*Honkoop et al., 2019*). Moreover, we treated zebrafish embryos with an Erbb2 inhibitor (*Figure 2e*), which has been shown to severely affect trabeculation (*Figure 2—figure supplement 1b*) (*Liu et al., 2010*; *Peshkovsky et al., 2011*), and found that the cardiac expression

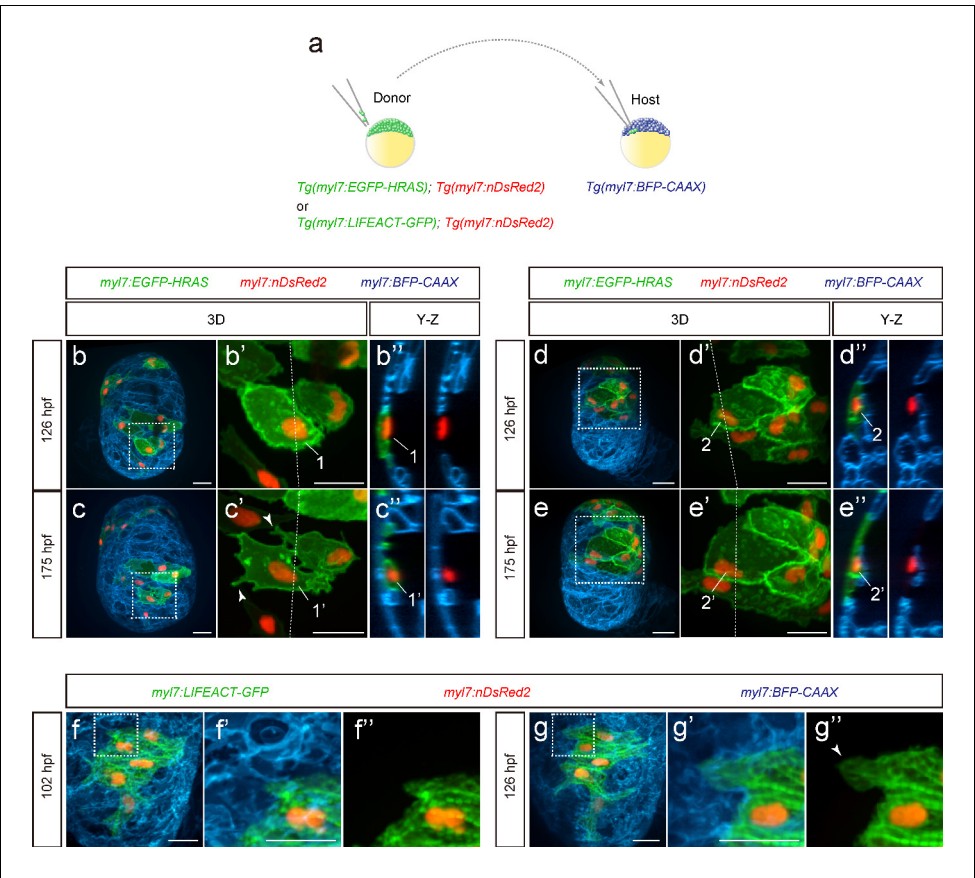

**Figure 1.** Cardiomyocyte behavior during cardiac trabeculation. (**a**) Schematic of the transplantation experiment. (**b–e**) 3D time-course images of chimeric hearts; magnified view (**b', c', d', e'**) of area in white boxes and Y-Z plane images (**b", c", d", e"**) along white dashed lines (**b', c', d', e'**). CMs initially in the compact layer (**b', b"**) enter the trabecular layer (**c', c"**) exhibiting morphological changes and membrane protrusions (**c'**; arrowheads; *n* = 5 CMs); CMs remaining in the compact layer (**d', d", e', e"**) do not exhibit obvious morphological changes (*n* = 5 CMs). The same CMs are shown at 126 and 175 hpf as indicated in the images. (**f, g**) 3D time-course images of chimeric heart; magnified view (**f', f", g', g"**) of area in white boxes. CMs entering the trabecular layer exhibit partial disassembly of their sarcomeres (**g'**; arrowhead). Scale bars, 20 μm.

The online version of this article includes the following video and figure supplement(s) for figure 1:

**Figure supplement 1.** Cardiomyocytes change morphology and exhibit cell-cell junction rearrangements when entering the trabecular layer.

**Figure 1—video 1.** Z-stack images of 126 hpf CMs related to *Figure 1b'*.
https://elifesciences.org/articles/50161#fig1video1

**Figure 1—video 2.** Z-stack images of 175 hpf CMs related to *Figure 1c'*.
https://elifesciences.org/articles/50161#fig1video2

---

of glycolytic enzyme genes was downregulated (*Figure 2b*). Altogether, these data indicate that ERBB2 signaling activates glycolysis in CMs.

## Glycolysis regulates cardiomyocyte delamination during development

In order to analyze the role of glycolysis during trabeculation, we first focused on pyruvate metabolism. The pyruvate dehydrogenase complex (PDC) catalyzes the conversion of pyruvate to acetyl-coenzyme A (acetyl-CoA), which enters the tricarboxylic acid cycle (*Zhang et al., 2014*). Pyruvate dehydrogenase kinases (PDKs) inhibit PDC activity and enhance glycolysis in CMs (*Zhao et al., 2008*) and cancer cells (*Koukourakis et al., 2005*; *Lu et al., 2008*; *Leclerc et al., 2017*; *Peng et al., 2018*), thereby regulating the switch between glycolysis and OXPHOS (*Zhang et al., 2014*). Our analyses show that ERBB2 signaling positively regulates *Pdk3* gene as well as protein expression

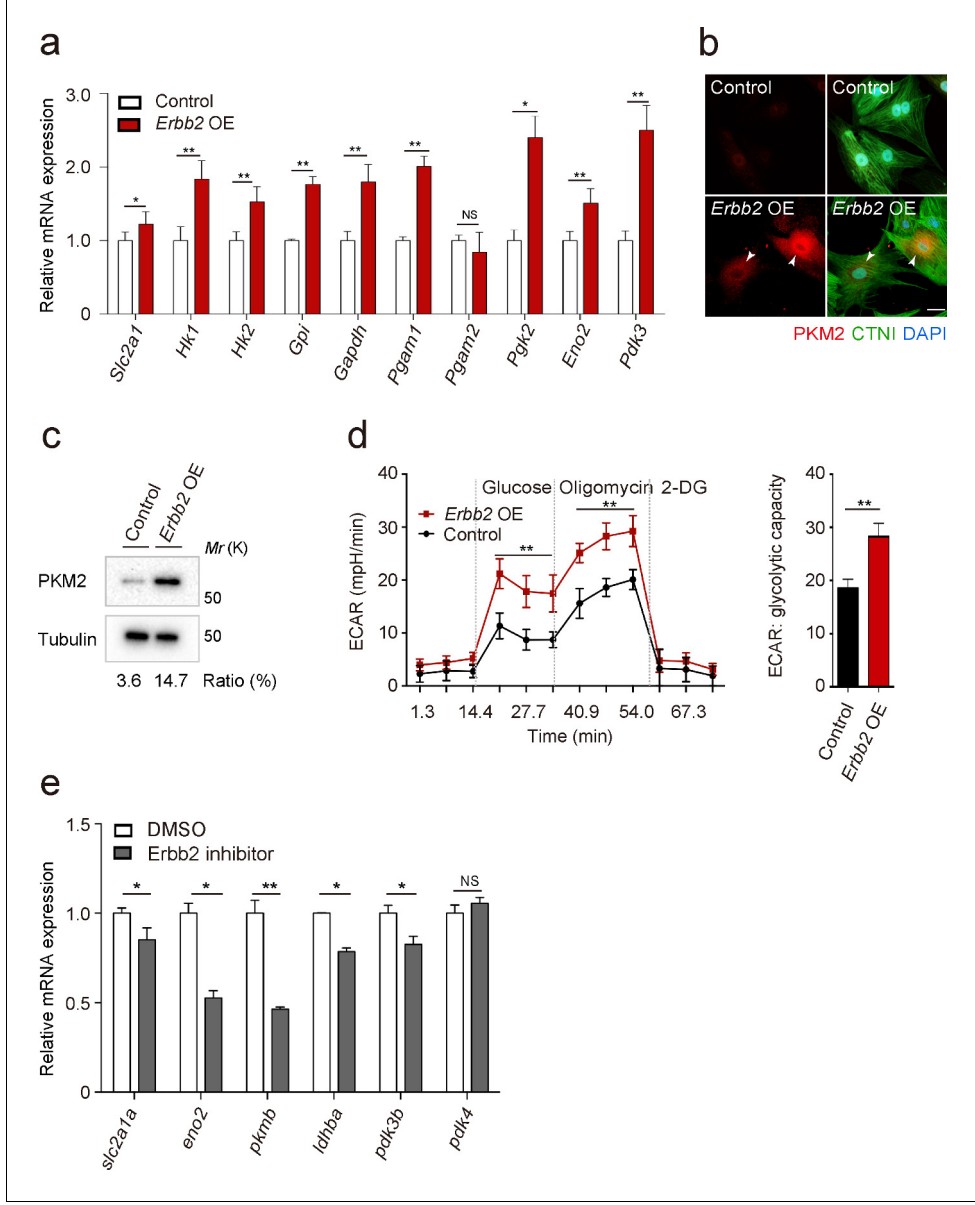

**Figure 2.** ERBB2 signaling activates glycolysis in cardiomyocytes. (**a**) qPCR analysis of mRNA levels of glycolytic enzyme genes in control and *Erbb2* overexpressing (OE) rat neonatal CMs (*n* = 3). Error bars, s.e.m. (**b**) Staining for PKM2, CTNI and DNA (DAPI) in control and *Erbb2* OE rat neonatal CMs; arrowheads point to PKM2+ CMs. (**c**) Western blot analysis of PKM2 levels in control and *Erbb2* OE rat neonatal CMs. (**d**) Extracellular acidification rate (ECAR) analysis in control and *Erbb2* OE rat neonatal CMs; glycolytic capacity shown on the right (*n* = 7). Error bars, s.d. (**e**) qPCR analysis of mRNA levels of glycolytic enzyme genes in DMSO and Erbb2 inhibitor treated zebrafish hearts (*n* = 3). Error bars, s.e.m.; *p<0.05 and **p<0.001 by two-tailed unpaired *t*-test. NS, not significant. Scale bar, 20 μm.

The online version of this article includes the following source data and figure supplement(s) for figure 2:

**Source data 1.** Mass spectrometry data.
**Source data 2.** Primer sequences for qPCR analysis.
**Source data 3.** Mean Ct values of qPCR analysis in *Figure 2a and e*.
**Figure supplement 1.** NRG1/ERBB2 signaling activates glycolysis in CMs.
**Figure supplement 2.** Uncropped images related to western blotting data.

(*Figure 2a and b* and *Figure 2—source data 1*). Thus, we hypothesized that PDK3 was one of the key enzymes regulating glycolysis in delaminating CMs in response to Erbb2 signaling. Consistent with this model, we found that the PDK inhibitor dichloroacetate (DCA) led to a significant reduction in the number of CMs in the trabecular layer (*Figure 3a*). Of note, this phenotype is similar to the one caused by Erbb2 inhibition (*Figure 3a*).

We next focused on *pyruvate dehydrogenase E1 alpha 1 subunit a* (*pdha1a*), which encodes a catalytic subunit of the PDC. Analysis of CM-specific loss of *Pdha1* in mice has revealed the importance of PDC activity for OXPHOS (*Sun et al., 2016*). We generated a *Tg(myl7:pdha1aSTA-P2A-tdTomato)* line to overexpress an activated form of Pdha1a in CMs. This activated form of Pdha1a (Pdha1aSTA) contains mutations in its phosphorylation sites and thus is not inhibited by PDK. As a result, glycolysis is reduced and OXPHOS enhanced, as previously shown in cancer cells (*Hitosugi et al., 2011*; *Fan et al., 2014*). Notably, larvae expressing this activated form of Pdha1a exhibited a significant decrease in the number of CMs in the trabecular layer (*Figure 3b* and *Figure 3—figure supplement 1a*). We also generated a *Tg(myl7:pdk3b-P2A-tdTomato)* line to overexpress *pdk3b* in CMs and thereby promotes glycolysis (*Lu et al., 2008*), and found that these transgenic larvae exhibited a significant increase in the number of CMs in the trabecular layer (*Figure 3b*). Next, we tested whether the modulation of glycolysis affected CM proliferation and found that *Tg(myl7:pdha1aSTA-P2A-tdTomato)* or *Tg(myl7:pdk3b-P2A-tdTomato)* larvae did not exhibit a significant change compared to WT in the percentage of mVenus-gmnn+ CMs (*Figure 3—figure supplement 1b*). Furthermore, we examined whether the modulation of glycolysis affected CM morphology in rat neonatal CMs in culture and found that *Pdk3* overexpression led to the induction of membrane protrusions (*Figure 3d and d'*), as well as cell-cell junction rearrangements (*Figure 3d–d"*). Similar effects were also observed following *Erbb2* overexpression (*Figure 3e–e"*). Together, these data suggest that CM morphological changes regulated by glycolysis are important for delamination.

In order to further assess the role of glycolysis in trabeculation, we focused on zebrafish *pkm2* to analyze a glycolytic enzyme mutant model. Loss of *PKM2* has been shown to impair glycolysis in endothelial cells (*Stone et al., 2018*), and PKM2 expression has been associated with glycolysis and cell growth in cancer cells (*Christofk et al., 2008*). Moreover, in rat neonatal CMs, *Erbb2* OE upregulated *Pkm2* (*Figure 2b and c*). Mammalian *Pkm* encodes two splice variants (M1 and M2 isoforms); PKM2 plays an important role in glycolysis, while PKM1 promotes OXPHOS (*Christofk et al., 2008*; *Lunt et al., 2015*; *Zheng et al., 2016*). Zebrafish *pkma2*, a splice variant of *pkma*, and *pkmb* are the orthologues of mammalian *Pkm2* (*Stone et al., 2018*). During early development, *pkma* is expressed in the heart, head, spinal cord and blood vessels, while *pkmb* is highly expressed in the somites (*Figure 4—figure supplement 1a*). At later stages, *pkmb* becomes clearly expressed in the heart (*Figure 4—figure supplement 1b*; *Gunawan et al., 2019*). We examined *pkma2*; *pkmb* double mutants in which Pkma1, which drives pyruvate metabolism via OXPHOS, remains intact, and found that loss of *pkma2* and *pkmb* impaired trabeculation (*Figure 4—figure supplement 1c*). We did not find evidence for increased CM apoptosis in *pkma2*; *pkmb* double mutants compared to WT (*Figure 4—figure supplement 1d*). In order to examine the CM-specific role of *pkma2* and *pkmb* in trabeculation, we performed cell transplantation experiments whereby *pkma2*; *pkmb* double heterozygous and double mutant cells were transplanted into WT embryos (*Figure 4a*). We found a significantly lower percentage of double mutant versus double heterozygous CMs in the trabecular layer of mosaic hearts (*Figure 4b–c*), indicating the importance of these genes in trabeculation. We also counted the number of trabecular CMs in these chimeric hearts and observed no significant deviation from WT (*Figure 4—figure supplement 1e*). Altogether, these results indicate that glycolysis plays important and CM-autonomous roles during trabeculation.

## Discussion

During trabeculation, CMs exhibit membrane protrusions (*Staudt et al., 2014*) and rearrange their cell-cell junctions (*Cherian et al., 2016*; *Miao et al., 2019*). Our 3D single CM imaging clearly reveals that CMs that enter the trabecular layer change their shape, similar to migrating cells, and lose cell-cell adhesion, indicating that they undergo phenotypic changes. Epithelial cells exhibit cellular plasticity as they change shape, and lose cell-cell adhesion and apicobasal polarity - a phenotypic transformation called epithelial to mesenchymal transition (EMT) (*Nieto, 2013*; *Ye and Weinberg, 2015*; *Varga and Greten, 2017*). Recent studies suggest that endothelial cells can also undergo

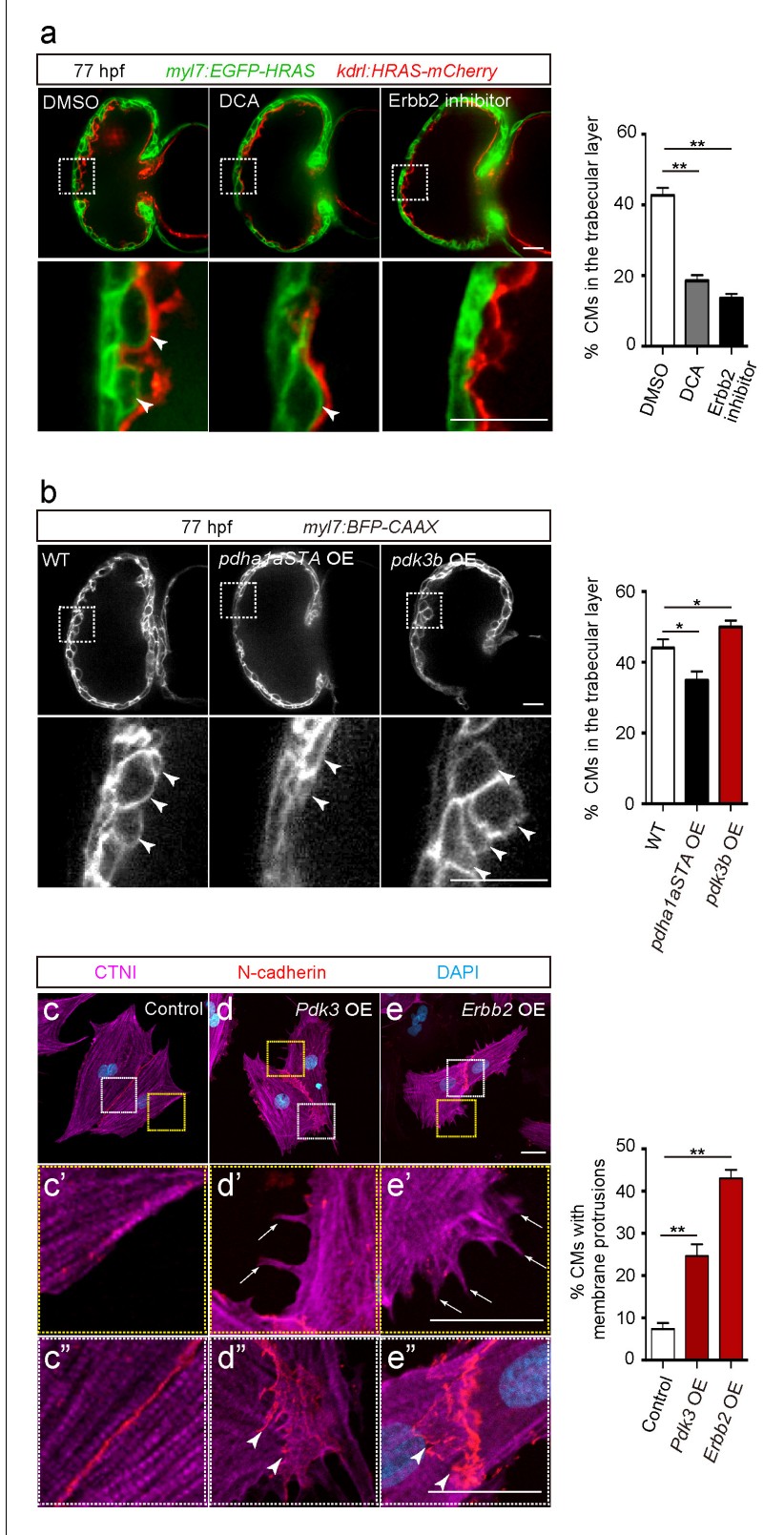

**Figure 3.** Glycolysis regulates cardiac trabeculation. (a) Confocal images (mid-sagittal sections) of 77 hpf hearts treated with DMSO, dichloroacetate (DCA) or Erbb2 inhibitor; magnified view of area in white boxes shown below; arrowheads point to CMs in the trabecular layer; percentage of CMs in the trabecular layer shown on the right (*n* = 5–7 ventricles). (b) Confocal images (mid-sagittal sections) of 77 hpf *Tg(myl7:BFP-CAAX)* alone or in

*Figure 3 continued on next page*

*Figure 3 continued*

combination with *Tg(myl7:pdha1aSTA-P2A-tdTomato)* or *Tg(myl7:pdk3b-P2A-tdTomato)* hearts; magnified view of area in white boxes shown below; arrowheads point to CMs in the trabecular layer; percentage of CMs in the trabecular layer shown on the right (*n* = 5–7 ventricles). (**c–e"**) Staining for CTNI, N-cadherin and DNA (DAPI) in control (**c**), *Pdk3* (**d**) and *Erbb2* (**e**) OE rat neonatal CMs; magnified view of area in yellow (**c', d', e'**) and white (**c", d", e"**) boxes; percentage of CMs exhibiting membrane protrusions shown on the right (*n* = 3 individual experiments; each value corresponds to an average of 30 CMs). *Pdk3* and *Erbb2* OE causes rat neonatal CMs to exhibit membrane protrusions (**d', e'**; arrows) and cell-cell junction rearrangements (**d', e'**; arrowheads). Error bars, s.e.m.; *p<0.05 and **p<0.001 by ANOVA followed by Tukey's HSD test. Scale bars, 20 μm.

The online version of this article includes the following figure supplement(s) for figure 3:

**Figure supplement 1.** Cardiomyocyte proliferation does not appear to be affected by modulation of glycolysis.

phenotypic changes towards mesenchymal-like cells (*Markwald et al., 1977*; *Zeisberg et al., 2007*; *Pearson, 2015*; *Dejana et al., 2017*; *Kovacic et al., 2019*). Before the onset of trabeculation, compact layer CMs exhibit apicobasal polarity, and then some of them depolarize and subsequently delaminate to seed the trabecular layer (*Jiménez-Amilburu et al., 2016*). Notably, ERBB2 signaling, which is essential for trabeculation (*Lee et al., 1995*; *Liu et al., 2010*), induces EMT in breast cancer cells (*Carpenter et al., 2015*; *Ingthorsson et al., 2016*). Altogether, these data indicate that during trabeculation CMs undergo an EMT-like process triggered by ERBB2 signaling.

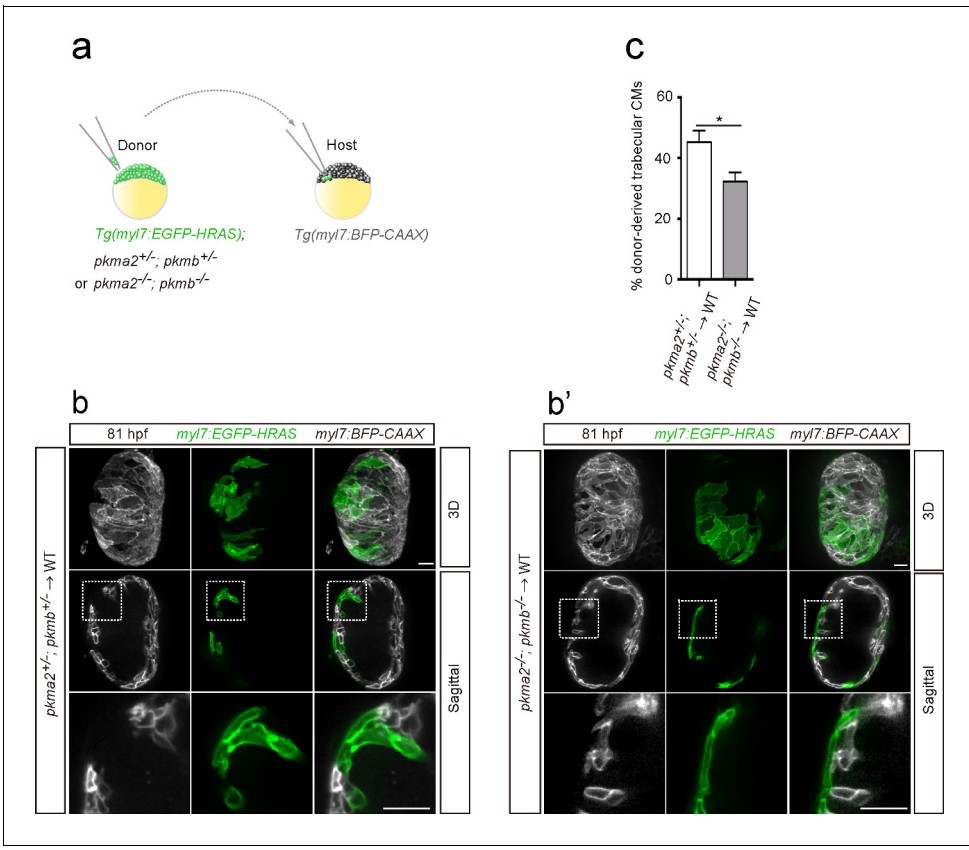

**Figure 4.** Loss of *pkm2* impairs cardiac trabeculation. (**a**) Schematic of the transplantation experiment. (**b, b'**) 3D and mid-sagittal section images of chimeric hearts using *pkma2⁺/⁻; pkmb⁺/⁻; Tg(myl7:EGFP-HRAS)* (**b**) and *pkma2⁻/⁻; pkmb⁻/⁻; Tg(myl7:EGFP-HRAS)* (**b'**) cells as donors; magnified view of area in white boxes shown below. (**c**) Percentage of donor-derived trabecular CMs (*n* = 10 ventricles). Error bars, s.e.m.; *p<0.05 by two-tailed unpaired *t*-test. Scale bars, 20 μm.

The online version of this article includes the following figure supplement(s) for figure 4:

**Figure supplement 1.** Glycolytic enzymes regulate cardiac trabeculation.

Cardiac metabolism is essential for energy production to sustain continuous cardiac contractions (*Doenst et al., 2013*). However, the role of metabolism during cardiac development remains unclear. Our study reveals that glycolysis regulates CM behavior during cardiac wall morphogenesis. Glycolysis enables rapid production of ATP to meet the high-energy demands of cell proliferation and migration in developing tissues, cancer cells and endothelial cells (*Lunt and Vander Heiden, 2011*; *Liberti and Locasale, 2016*; *Potente and Carmeliet, 2017*). Moreover, glycolytic intermediates are utilized to produce biomass including proteins and nucleic acids, further supporting these processes (*DeBerardinis et al., 2008*; *Potente and Carmeliet, 2017*). In addition, excessive glycolysis is associated with EMT in cancer cells (*Peppicelli et al., 2014*; *Huang and Zong, 2017*; *Morandi et al., 2017*). These findings indicate that glycolysis could regulate CM behavior in several different ways, and it will be interesting to dissect these processes further.

## Materials and methods

### Key resources table

| Reagent type (species) or resource | Designation | Source or reference | Identifiers | Additional information |
|---|---|---|---|---|
| Genetic reagent (*Danio rerio*) | *pkma2*[s717] | *Stone et al., 2018* | | |
| Genetic reagent (*Danio rerio*) | *pkmb*[s718] | *Stone et al., 2018* | | |
| Gene (*Danio rerio*) | *pdk3b* | | NM_001080688 | |
| Gene (*Danio rerio*) | *pdha1a* | | NM_213393 | |
| Gene (*Rattus norvegicus*) | *Pdk3* | | NM_001106581 | |
| Gene (*Homo sapiens*) | *ERBB2* | | NM_004448 | |
| Antibody | anti-Cardiac Troponin I (Goat polyclonal) | Abcam | AB_880622 Cat# ab56357 | IF(1:500) |
| Antibody | anti-PKM2 (Rabbit monoclonal) | Cell Signaling | AB_1904096 Cat# D78A4 | IF(1:100) WB(1:1000) |
| Chemical compound, drug | Sodium dichloroacetate | Sigma Aldrich | 347795 | |
| Software, algorithm | Zen 2012 (Blue edition) | Carl Zeiss Microscopy | Version 1.1.2.0 | |
| Software, algorithm | Imaris x64 | Bitplane | Version 9.3.0 | |
| Other | DAPI stain | Sigma | D954 | (1 µg/mL) |

### Zebrafish

All zebrafish husbandry was performed under standard conditions in accordance with institutional (MPG) and national ethical and animal welfare guidelines. The following transgenic lines and mutants were used: *Tg(myl7:EGFP-Has.HRAS)*[s883] (*D'Amico et al., 2007*) abbreviated *Tg(myl7:EGFP-HRAS)*, *Tg(myl7:LIFEACT-GFP)*[s974] (*Reischauer et al., 2014*), *Tg(−5.1myl7:DsRed2-NLS)*[f2Tg] (*Mably et al., 2003*) abbreviated *Tg(myl7:nDsRed2)*, *Tg(kdrl:Has.HRAS-mCherry)*[s896] (*Chi et al., 2008*) abbreviated *Tg(kdrl:HRAS-mCherry)*, *Tg(myl7:cdh2-tdTomato)*[bns78] (*Fukuda et al., 2017*), *Tg(myl7:BFP-CAAX)*[bns193] (*Guerra et al., 2018*), *Tg(myl7:mVenus-Gemnn)*[ncv43Tg] (*Jiménez-Amilburu et al., 2016*), *pkma2*[s717] (*Stone et al., 2018*) and *pkmb*[s718] (*Stone et al., 2018*). To generate *Tg(myl7:pdk3b-P2A-tdTomato)*[bns365] and *Tg(myl7:pdha1aSTA-P2A-tdTomato)*[bns366], *pdk3b* (NM_001080688) and *pdha1a* (NM_213393) were isolated by RT-PCR and cloned under the control of the *myl7* promoter in a vector containing *Tol2* elements and two I-SceI restriction enzyme sites. The following primers were used to amplify the cDNA: *pdk3b* (forward 5'- AAGCAGACAGTGAACAAGCTTCCACCATGAAAC

TGTTTATCTGCCTACTG-3' and reverse 5'-TAGCTCCGCTTCCGTCGACTCTGTTGACTTTGTATG TGGAC-3'); *pdha1a* (forward 5'-AAGCAGACAGTGAACAAGCTTCCACCATGAGAAAGATGC TAACCATAATT-3' and reverse 5'-TAGCTCCGCTTCCGTCGACGCTGATGGACTTGAGTTTG-3').

To generate the plasmid encoding an activated form of *pdha1a* (*pdha1aSTA*), the equivalent residues for human PDHA1 Ser293 and Tyr301 were replaced by alanine using the following primers: *pdha1aSTA* (forward 5'- CTATCGTTATCATGGACACGCTATGAGCGACCCAGGAGTCAGCGCCCG-CACACGTGAGGAGA-3' and reverse 5'- TTCCCTCACGTGTGCGGGCGCTGACTCCTGGGTCGC TCATAGCGTGTCCATGATAACGATAG-3'). Plasmids were then injected into one-cell stage embryos with I-SceI (NEB) or *Tol2* mRNA.

## Quantification of CMs in the trabecular layer

Quantification of trabecular CMs in 77 and 81 hpf heats was performed as previously described (*Jiménez-Amilburu et al., 2016*) using the ZEN software (ZEISS). Starting from the mid-sagittal plane, we quantified trabecular versus compact layer CMs in the ventricular outer curvature, 12 planes up and 12 planes down at an increment of 1 µm per plane. To quantify donor-derived trabecular CMs in chimeric hearts generated by cell transplantation, we counted the number of donor-derived trabecular and compact layer CMs. Then, the percentage of donor-derived trabecular CMs was calculated by dividing the number of donor-derived trabecular CMs by the total number of donor-derived CMs. Quantification of trabecular CMs in 131 hpf hearts was performed using the ZEN software (ZEISS). Starting from the mid-sagittal plane, we measured the whole myocardial area as well as the trabecular area, 10 µm up and 10 µm down. Three different sagittal planes per heart were measured. The percentage of the trabecular area for each sagittal plane was calculated by dividing the trabecular area by the myocardial area, and the average value was used for the graph.

## In situ hybridization

In situ hybridization was performed as previously described (*Thisse and Thisse, 2008*). To synthesize *pkma* (NM_199333) and *pkmb* (NM_001003488) RNA probes, the following primers were used to amplify the corresponding DNA fragments: *pkma* (forward 5'- TTGGATCCACCATGTCTCAAAC TAAAGCTC-3' and reverse 5'- TTTGAATTCTTACGGCACTGGGACGACAC-3'); *pkmb* (forward 5'-TTGGATCCACCATGTCTCAGACAAAGACTA-3' and reverse 5'- TTTGAATTCTCAAGGCACCA-CAACGATG'). The DNA fragments were cloned into the pCS2 vector. DIG-labeled RNA probes were synthesized using a DIG RNA labeling kit (Sigma-Aldrich) and MegaScript T7 Transcription Kit (Thermo Fisher Scientific).

## Pharmacological treatments

Zebrafish embryos were treated with DMSO (control), 30 mM DCA (Sigma-Aldrich) or 5 µM Erbb2 inhibitor (AG1478; Sigma-Aldrich) from 50 hpf to 77 hpf and then analyzed.

## TUNEL assay

To examine apoptosis, an in situ cell death detection kit (Roche) was used.

## Immunostaining

Rat neonatal CMs were fixed in 4% paraformaldehyde. Anti-cardiac troponin I (CTNI) 1:500 (ab56357, Abcam) and anti-PKM2 1:100 (D78A4, Cell Signaling) were used. After washing with PBS, samples were stained with Alexa-568, Alexa-488 or Alexa-647 secondary antibodies 1:500 (Life Technologies), followed by 4',6-Diamidine-2'-phenylindole dihydrochloride (DAPI) 1:2000 (Merck) staining to visualize DNA.

## Cell culture

Rat neonatal (P2-P4) CMs were isolated and cultured as previously described (*Fukuda et al., 2017*). Cells were plated onto 0.1% gelatin-coated (Sigma) plates and cultured in DMEM/F12 (Gibco) supplemented with 5% horse serum, L-glutamine, Na-pyruvate, penicillin and streptomycin at 37°C and 5% $CO_2$. Adenovirus vectors for transfection into CMs were generated using the AdEasy system (Agilent Technologies). To generate adenovirus vectors encoding PDK3 (NM_001106581) or ERBB2 (NM_004448), the following primers were used to amplify *Pdk3* from rat neonatal CM or *ERBB2*

from Addgene clone # 39321: *Pdk3* (forward 5'-TAGAGATCTGGTACCGTCGACCACCATGCGGC
TCTTCTACCG-3' and reverse 5'-GGATATCTTATCTAGAAGCTTCTAGAAAGTTTTATTACTCTTGATC
TTGTCC-3'); *ERBB2* (forward 5'-TAGAGATCTGGTACCGTCGACGCGGCCGCACCACCATGTA
TCCATATGATGTTCCAGATTATGCTATGGAGCTGGCGGCCTTG-3' and reverse 5'-GGATATCTTA
TCTAGAAGCTTTCACACTGGCACGTCCAG).

## Imaging

Zebrafish embryos and larvae were anesthetized with 0.2% tricaine and mounted in 1% low-melting agarose. A Zeiss spinning disk confocal microscope system (CSU-X1, Yokogawa) and ORCA-flash4.0 sCMOS camera (Hamamatsu) was used to acquire images. 3D images were processed using Imaris (Bitplane). Circularity was measured using ImageJ (NIH).

## Western blotting

Protein expression levels were analyzed by western blotting as previously described (*Fukuda et al., 2017*). In brief, proteins were extracted with lysis buffer (150 mM Tris-HCl pH 7.5, 150 mM NaCl, 1% Triton X-100, 0.2% SDS, 1 mM EDTA, 5 mM NaF, 0.1 mM orthovanadate, 1 mM phenylmethylsulfonyl fluoride and 1 μg/ml aprotinin). Proteins were separated by SDS-PAGE. The following primary antibodies were used: anti-PKM2 1:1000 (D78A4, Cell Signaling) and anti-alpha-Tubulin 1:1000 (T6199, Sigma).

## qPCR

Rat neonatal CMs were transfected with adenovirus vectors encoding genes of interest. 24 hr after transfection, total RNA was extracted. Zebrafish embryos and larvae were treated with DMSO or Erbb2 inhibitor from 55 to 106 hpf, and then the hearts were isolated to extract total RNA. A miR-Neasy Mini kit (Qiagen) was used for total RNA extraction and cDNA was synthesized using a SuperScript Second Strand kit (Life Technologies). A CFX Connect Real-Time system (Bio-Rad) and DyNAmo colorFlash SYBR green qPCR kit (ThermoFisher Scientific) were used. Primer sequences are shown in *Figure 2—source data 2* and Ct values in *Figure 2—source data 3*.

## Metabolic assays

The extracellular acidification rate (ECAR) was measured using a Seahorse XFe96 analyzer (Seahorse Bioscience) following manufacturer's protocol. Rat neonatal CMs were seeded at 30,000 cells/well on to 0.1% gelatin-coated XFe96 microplates (Agilent Technologies) in DMEM/F12 (Gibco) containing 10% fetal bovine serum (FBS), L-glutamine, Na-pyruvate, penicillin and streptomycin at 37°C and 5% $CO_2$. After 24 hr of culture, the medium was replaced with serum-free medium. Then, cells were transfected with adenovirus vectors encoding genes of interest or a mock adenovirus vector, or treated with NRG1 (100 ng/ml; Abcam). 24 hr after transfection or NRG1 treatment, cells were maintained in non-buffered assay medium (Agilent Technologies) in a non-$CO_2$ incubator 1 hr prior to the assay. A glycolysis stress test kit (Seahorse Bioscience) was used to monitor ECAR where baseline measurements were made followed by sequential injection of glucose (10 mM), oligomycin (2 μM), and 2-DG (100 mM).

## Acknowledgements

We thank Beate Grohmann, Radhan Ramadass, Srinath Ramkumar, Simon Perathoner, Carmen Büttner, Nana Fukuda and Sharon Meaney-Gardian for help and support, Arica Beisaw, Ruben Marin-Juez, Josephine Gollin and Rashmi Priya for comments on the manuscript, and Jeroen Bakkers for communication. This work was supported in part by funds from the Max Planck Society to DYRS.

## Additional information

### Competing interests

Didier YR Stainer: Senior editor, *eLife*. The other authors declare that no competing interests exist.

## Funding

| Funder | Grant reference number | Author |
|---|---|---|
| Max-Planck-Gesellschaft | Open-access funding | Didier YR Stainier |

The funders had no role in study design, data collection and interpretation, or the decision to submit the work for publication.

## Author contributions

Ryuichi Fukuda, Conceptualization, Project administration; Alla Aharonov, Oliver A Stone, Mohamed El-Brolosy, Michael Potente, Resources, Investigation; Yu Ting Ong, Hans-Martin Maischein, Investigation; Eldad Tzahor, Resources; Didier YR Stainier, Supervision

## Author ORCIDs

Ryuichi Fukuda (iD) https://orcid.org/0000-0002-0281-5161
Yu Ting Ong (iD) http://orcid.org/0000-0003-3407-2515
Eldad Tzahor (iD) http://orcid.org/0000-0002-5212-9426
Didier YR Stainier (iD) https://orcid.org/0000-0002-0382-0026

## Ethics

Animal experimentation: All zebrafish husbandry was performed under standard conditions in accordance with institutional (MPG) and national ethical and animal welfare guidelines approved by the ethics committee for animal experiments at the Regional Board of Darmstadt, Germany (permit numbers B2/1017, B2/1041 and B2/1159).

## Decision letter and Author response

Decision letter https://doi.org/10.7554/eLife.50161.sa1
Author response https://doi.org/10.7554/eLife.50161.sa2

# Additional files

## Supplementary files

• Transparent reporting form

## Data availability

All data in this study are included in the manuscript and supporting files.

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
