## [Decision Letter]

**Acceptance summary:**

This is an elegant study that asks multiple questions about the roles of metabolism in cardiac wall formation and the initiation of trabeculation. Cell transplantations between different genetic/transgenic backgrounds allows detailed analysis of the behavior of cells at the initiation of trabeculation. Overexpression of constitutively active ERRB2 (OE) resulted in upregulation of glycolytic pathway and activity in mouse transgenics, rat neonatal cardiomyocyte cultures, and downregulation of oxidative phosphorylation pathways; the converse was seen with inhibition of Errb2. Multiple approaches, including cardiomyocyte-specific expression of transgenics in zebrafish and *pkma2; pkmb* double mutants, indicate the importance of glycolysis for initiation of trabeculation, downstream of Errb2.

**Decision letter after peer review:**

Thank you for submitting your article "Metabolic modulation regulates cardiac wall morphogenesis and regeneration in zebrafish" for consideration by *eLife*. Your article has been reviewed by three peer reviewers, and the evaluation has been overseen by Marianne Bronner as the Senior and Reviewing Editor. The following individual involved in review of your submission has agreed to reveal their identity: Shawn Burgess (Reviewer #1).

The reviewers have discussed the reviews with one another and the Reviewing Editor has drafted this decision to help you prepare a revised submission.

Although all of the reviewers are enthusiastic about the manuscript, they concur that it would be important to drop the regeneration component of the manuscript, which feels a bit added on and is not entirely convincing. Other changes are recommended but non-essential. The full reviews are included below for your information.

Reviewer #1:

Fukuda and colleagues have submitted a manuscript describing the role of metabolic shifts in cardiac development and regeneration. The majority of the work focuses on a particular developmental stage where the inner walls of the heart begin forming from the compact layer via delaminating cardiomyocytes (trabeculation). ErbB2 signaling has been shown to be important for this process and modulation of erbb2 activity levels is the starting point for this manuscript. Across three model systems, increases in ErbB2 activity increases genes related to glycolysis and decreases in ErbB2 signaling have matching decreases in genes involved in glycolysis. The authors targeted specific enzymes modulating pyruvate metabolism, a summary of the results are 1) inhibiting pyruvate dehydrogenase kinase decreases CMs, 2) increasing PDK activity increases CMs, 3) increasing pyruvate dehydrogenase activity decreases CMs, 4) decreasing pyruvate kinase decreases CMs. Shifting attention to cardiac regeneration in adult zebrafish, similar roles for metabolic shifts are seen, inhibition of PDK (with DCA) or phosphoglucose isomerase (2DG) reduces regeneration as does loss of PK activity while loss of ppargc1a (involved in mitochondrial biosynthesis) stimulates regeneration. In general, these are well-controlled experiments that add significantly to our understanding of the relationship between metabolic modulation and embryonic morphogenesis.

There are some problems with conclusions and/or conflicting data that should be addressed.

1) The authors have focused primarily on enzymes involved in the last steps of glycolysis: the pyruvate dehydrogenase complex (PDC), PDK, and pyruvate kinase (PK). It is true that these enzymes are listed in the glycolysis pathway, but except for PK which does release one ATP molecule, the other enzymes are primarily involved in shunting pyruvate either towards lactic acid (and the pentose phosphate pathway) or into the TCA cycle. This shift point isn't really modulating glycolysis but the choice between aerobic and anaerobic respiration and it may be that pyruvate concentration and/or utilization is the key step they are looking at and not glycolysis per se. I am arguing this point because otherwise the loss of PK activity (*pkma2/pkmb*) resulting in less trabeculation and regeneration appears to conflict with the other observations. Increasing PDK activity pushes pyruvate towards lactate (shunting away from mitochondria) resulting in an increase in trabeculation, blocking PDK activity has the opposite effect, less trabeculation more TCA cycle. The dominant active pyruvate dehydrogenase experiments are also consistent with these results, more PDK activity = less trabeculation. The PK knockouts should also reduce the availability of pyruvate for the mitochondria, yet it results in less trabeculation and less regeneration. Perhaps lactate is a key signaling molecule? This has been shown in neurons for example.

2) I think the authors need to change the general use of glycolysis throughout the manuscript to specify whether they are talking about anaerobic or aerobic glycolysis. Can the redox state be determined during trabeculation?

Reviewer #2:

There are a few questions about the trabeculation results, but overall this part of the study is convincing and an important contribution to the heart development field. In contrast, there are significant concerns about the heart regeneration aspects of the paper.

1) The interpretation that cardiac regeneration is altered is not convincing. The assays for regeneration were only taken to 5 days post cryo injury (dpci), which is not sufficient time to assess whether regeneration is normal or abnormal. The hearts in any of the experimental cohorts could be slightly delayed and then fully recovered by 30 dpci. There is an additional concern that the recovering fish are heat shocked every day for the first five days up to analysis. The standard in the field is 30- or 60-days post injury, with intermittent time points. I suggest removing the regeneration aspect of this paper. It is just too premature, unsupported conclusions in the title and Abstract could be incorrect.

2) In Figure 4, cell transplantation is used to generate chimeric hearts is a very useful experiment. I imagine the numbers are in hand, but it would be useful to see them analyzed in a few different ways. Are the total number of trabecular cardiomyocytes decreased in the hearts that received mutant donor cells? Or is it just a shift between the mutant vs. host cells that can contribute to the trabeculae? Figure 4C indicates that the% of cardiomyocytes that are trabecular was decreased in hearts that received homozygous mutant cells. However, it does not tell us whether the cardiomyocytes that are trabecular are from the transplanted or host cells. The legend says 'percentage of donor CMs' However, Figure 4B' appears to show some labeled (transplant) cells that are interior (trabecular), outside the white box. It would be useful to know the number of labeled (transplant) vs. unlabeled (host) cells that are trabecular, to understand whether the effect is cell autonomous or non-autonomous.

3) In Supplementary Figure 5, it appears that mCherry is also expressed in the epicardium after TAM and HS treatment. If this is the case, interpretation of results will need modification.

Reviewer #3:

In this manuscript, Fukuda et al. describe a role for glucose metabolism in not only normal development of the trabeculated myocardium, but also in the successful regeneration of the adult heart. Much is known about metabolic processes in the adult heart; here, the authors present a previously understudied role of glycolysis during development. First, by chimeric analysis to produce a higher-resolution look by live imaging, the authors show that cardiomyocytes elicit changes in cell shape to properly develop a trabeculated myocardium. Furthermore, the authors show Nrg1/Erbb2 signaling is required for trabeculation and activates glycolysis, which regulates development of the trabecular myocardium. Finally, the authors extend their study to adulthood and show that glycolysis is essential for successful dedifferentiation and proliferation of cardiomyocytes in cryoinjured adult hearts by upregulation of glycolytic enzymes at the wound border after injury. Overall, the authors contribute a new understanding of the role of glycolysis in proper heart development and repair.

1) The authors describe cell shape changes and membrane protrusions of cardiomyocytes to enter the trabecular layer. While the images are that of live embryos, this argument would be strengthened by time-lapse imaging to track the individual cardiomyocytes changing shape and position over time.

2) Methods used to determine glycolytic activity by assaying extracellular acidification rate should be explained in more detail.

3) The authors use transplanted double heterozygous and double mutant cells to show double mutant cells cannot incorporate into trabeculae. While these are very careful experiments, the health of these cells after transplantation is a concern. Have the authors scored transplant survival? Or performed a TUNEL assay to ensure the mutant cells are not dying after transplantation? Is delamination the only cellular event (cell motility, shape change, membrane protrusions, etc) that is perturbed?

4) Ppargc1a regulates genes involved in OXPHOS and are absent from regenerating hearts, but are they present in uninjured hearts by in situ?

---

## [Author Response]

Reviewer #1:

[…] There are some problems with conclusions and/or conflicting data that should be addressed.

1) The authors have focused primarily on enzymes involved in the last steps of glycolysis: the pyruvate dehydrogenase complex (PDC), PDK, and pyruvate kinase (PK). It is true that these enzymes are listed in the glycolysis pathway, but except for PK which does release one ATP molecule, the other enzymes are primarily involved in shunting pyruvate either towards lactic acid (and the pentose phosphate pathway) or into the TCA cycle. This shift point isn't really modulating glycolysis but the choice between aerobic and anaerobic respiration and it may be that pyruvate concentration and/or utilization is the key step they are looking at and not glycolysis per se. I am arguing this point because otherwise the loss of PK activity (pkma2/pkmb) resulting in less trabeculation and regeneration appears to conflict with the other observations. Increasing PDK activity pushes pyruvate towards lactate (shunting away from mitochondria) resulting in an increase in trabeculation, blocking PDK activity has the opposite effect, less trabeculation more TCA cycle. The dominant active pyruvate dehydrogenase experiments are also consistent with these results, more PDK activity = less trabeculation. The PK knockouts should also reduce the availability of pyruvate for the mitochondria, yet it results in less trabeculation and less regeneration. Perhaps lactate is a key signaling molecule? This has been shown in neurons for example.

The mammalian *Pkm* gene gives rise to two splice isoforms, *Pkm1* and *Pkm2*. The zebrafish genome contains two *pkm* genes, *pkma* and *pkmb. pkma* encodes both an orthologue of mammalian *Pkm1 (pkma1*) and an orthologue of mammalian *Pkm2 (pkma2*), while *pkmb* encodes only another orthologue of mammalian *Pkm2* (Stone et al., 2018).

Expression of PKM2 is associated with metabolism of pyruvate to lactate. Loss of PKM2 has been shown to result in compensatory upregulation of PKM1, which drives enhanced mitochondrial metabolism of pyruvate (Christofk et al., 2008; Lunt et al., 2015; Zheng et al., 2016). Therefore, with respect to pyruvate metabolism, the loss of PKM2 mimics blocking PDK activity, whereas the loss of PKM1 mimics enhancing PDK activity. In *pkma2; pkmb* double mutant zebrafish, the *pkma1* isoform, which drives pyruvate metabolism via OXPHOS, remains intact. Thus, in *pkma2; pkmb* double mutant zebrafish we expect a reduction in the metabolism of pyruvate to lactate and an increase in pyruvate oxidation, similar to the situation observed following inhibition of PDK activity. These findings are consistent with the other models that have been analyzed (Christofk et al., 2008; Zheng et al., 2016).

The revised sentence reads as follows:

‘We examined *pkma2; pkmb* double mutants in which Pkma1, which drives pyruvate metabolism via OXPHOS, remains intact, and found that loss of *pkma2* and *pkmb* impaired trabeculation (Figure 4—figure supplement 1C).’

2) I think the authors need to change the general use of glycolysis throughout the manuscript to specify whether they are talking about anaerobic or aerobic glycolysis.

Although PKM2 (Christofk et al., 2008) and PDK (Takubo et al., 2013) play an important role during aerobic glycolysis (Vander Heiden et al., 2009; Lunt and Vander Heiden, 2011), we did not assess oxygen levels in our models, and thus prefer to use “glycolysis”.

Can the redox state be determined during trabeculation?

We have now examined the redox state using the cardiomyocyte specific reporter line previously used to examine hydrogen peroxide levels (Panieri et al., 2017). We found no significant difference in reporter activity between compact and trabecular cardiomyocytes during delamination (Author response image 1).

**Author response image 1. respfig1:** Redox analysis. 70 hpf *Tg(myl7:mitochondrial-Rogofp2Orp1)* hearts were examined and 405 nm/488 nm ratios determined for redox state analysis. Relative values of 405 nm/488 nm in compact and trabecular cardiomyocytes are shown.

Reviewer #2:

There are a few questions about the trabeculation results, but overall this part of the study is convincing and an important contribution to the heart development field. In contrast, there are significant concerns about the heart regeneration aspects of the paper.1) The interpretation that cardiac regeneration is altered is not convincing. The assays for regeneration were only taken to 5 days post cryo injury (dpci), which is not sufficient time to assess whether regeneration is normal or abnormal. The hearts in any of the experimental cohorts could be slightly delayed and then fully recovered by 30 dpci. There is an additional concern that the recovering fish are heat shocked every day for the first five days up to analysis. The standard in the field is 30- or 60-days post injury, with intermittent time points. I suggest removing the regeneration aspect of this paper. It is just too premature, unsupported conclusions in the title and Abstract could be incorrect.

We have now removed the regeneration part from the revised manuscript.

2) In Figure 4, cell transplantation is used to generate chimeric hearts is a very useful experiment. I imagine the numbers are in hand, but it would be useful to see them analyzed in a few different ways. Are the total number of trabecular cardiomyocytes decreased in the hearts that received mutant donor cells? Or is it just a shift between the mutant vs. host cells that can contribute to the trabeculae? Figure 4C indicates that the% of cardiomyocytes that are trabecular was decreased in hearts that received homozygous mutant cells. However, it does not tell us whether the cardiomyocytes that are trabecular are from the transplanted or host cells. The legend says 'percentage of donor CMs.' However, Figure 4B' appears to show some labeled (transplant) cells that are interior (trabecular), outside the white box. It would be useful to know the number of labeled (transplant) vs. unlabeled (host) cells that are trabecular, to understand whether the effect is cell autonomous or non-autonomous.

We have now counted the total number of trabecular cardiomyocytes in the chimeric hearts and no significant differences were observed compared to WT (Figure 4—figure supplement 1D).

In Figure 4C, we counted the number of donor-derived cardiomyocytes in both the trabecular and compact layers of the chimeric hearts. We then calculated the percentage of donor-derived trabecular CMs by dividing the number of donor-derived trabecular CMs by the total number of donor-derived CMs. To clarify this point we have now modified Figure 4C, its legend and the relevant part of the Materials and methods section in the revised manuscript.

Together, our data indicate cardiomyocyte-autonomous function of *pkma2* and *pkmb* during trabeculation.

3) In Supplementary Figure 5, it appears that mCherry is also expressed in the epicardium after TAM and HS treatment. If this is the case, interpretation of results will need modification.

We have now removed the regeneration part from the revised manuscript.

Reviewer #3:

[…] 1) The authors describe cell shape changes and membrane protrusions of cardiomyocytes to enter the trabecular layer. While the images are that of live embryos, this argument would be strengthened by time-lapse imaging to track the individual cardiomyocytes changing shape and position over time.

Time-lapse imaging to track individual cardiomyocytes during trabeculation has been shown in a previous paper (Staudt et al., 2014). However, the 3D morphology of single cardiomyocytes that enter the trabecular layer remains mostly unclear, which is why we examined it.

2) Methods used to determine glycolytic activity by assaying extracellular acidification rate should be explained in more detail.

We have now modified the Materials and methods section in the revised manuscript to address the reviewer’s comment.

3) The authors use transplanted double heterozygous and double mutant cells to show double mutant cells cannot incorporate into trabeculae. While these are very careful experiments, the health of these cells after transplantation is a concern. Have the authors scored transplant survival? Or performed a TUNEL assay to ensure the mutant cells are not dying after transplantation?

We have now performed TUNEL assays and no elevated levels of apoptosis were detected in *pkma2; pkmb* mutant cardiomyocytes in chimeric hearts compared to WT cardiomyocytes (Figure 4—figure supplement 1D).

Is delamination the only cellular event (cell motility, shape change, membrane protrusions, etc) that is perturbed?

NRG/ERBB signaling also regulates cardiomyocyte proliferation (Zhao et al., 1998; Grego-bessa et al., 2007; Bersell et al., 2009; Rasouli and Stainier, 2017). We have now counted the number of proliferating cardiomyocytes during trabeculation and these data indicate that modulation of metabolism does not appear to affect cardiomyocyte proliferation (Figure 3—figure supplement 1B).

4) Ppargc1a regulates genes involved in OXPHOS and are absent from regenerating hearts, but are they present in uninjured hearts by in situ?

We have now removed the regeneration part from the revised manuscript.

**References**

Bersell, K., Arab, S., Haring, B., and Kühn, B. (2009). Neuregulin1/ErbB4 Signaling Induces Cardiomyocyte Proliferation and Repair of Heart Injury. Cell 138, 257–270.

Panieri, E., Millia, C., Santoro, M.M., (2017). Real-time quantification of subcellular H2O2 and glutathione redox potential in living cardiovascular tissues, Free Radic Bio Med, 109, 189-200.

Takubo, K., Nagamatsu, G., Kobayashi, C.I., Nakamura-Ishizu, A., Kobayashi, H., Ikeda, E., Goda, N., Rahimi, Y., Johnson, R.S., Soga, T., et al. (2013). Regulation of glycolysis by Pdk functions as a metabolic checkpoint for cell cycle quiescence in hematopoietic stem cells. Cell Stem Cell 12, 49–61.

Vander Heiden MG, Cantley LC, Thompson CB. (2009). Understanding the Warburg effect: the metabolic requirements of cell proliferation. Science 324:1029–33.

Zhao, Y.Y., Sawyer, D.R., Baliga, R.R., Opel, D.J., Han, X., Marchionni, M.A., and Kelly,

R.A. (1998). Neuregulins promote survival and growth of cardiac myocytes: Persistence of ErbB2 and ErbB4 expression in neonatal and adult ventricular myocytes. J. Biol. Chem. 273, 10261–10269.